# Private Multiparty Perception for Navigation

**Hui Lu, Mia Chiquier, Carl Vondrick**
Department of Computer Science
Columbia University
New York, NY 10027
{hl3231, mac2500, cv2428}@columbia.edu
https://visualmpc.cs.columbia.edu

## Abstract

We introduce a framework for navigating through cluttered environments by connecting multiple cameras together while simultaneously preserving privacy. Occlusions and obstacles in large environments are often challenging situations for navigation agents because the environment is not fully observable from a single camera view. Given multiple camera views of an environment, our approach learns to produce a multiview scene representation that can only be used for navigation, provably preventing one party from inferring anything beyond the output task. On a new navigation dataset that we will publicly release, experiments show that private multiparty representations allow navigation through complex scenes and around obstacles while jointly preserving privacy. Our approach scales to an arbitrary number of camera viewpoints. We believe developing visual representations that preserve privacy is increasingly important for many applications such as navigation.

## 1 Introduction

Navigation through cluttered environments is a fundamental challenge in computer vision and robotics, with many applications to autonomous vehicles and assistive technology. In the typical settings, agents receive a camera view of the surroundings and the task is to predict the sequence of actions needed to reach a goal destination (Figure 1 left). Today, many methods exist for solving this problem when the environments are fully observerable or predictable [9, 31, 37, 20], both in simulation [26] and the physical world [5, 13]. However, large environments often have many obstacles and occlusions, which create irreducible uncertainties that cannot be resolved from just a single input camera view.

Recently, the field has proposed different methods to combine multiple camera views together in order to create complete representations of scenes [31, 4, 32]. These methods use the geometry of cameras and visual correspondences between views to learn to reconstruct maps of environments, and they are highly efficient, scaling to the size of full neighborhoods [31]. Since multiview representations make the environment fully observerable, they allow for efficient navigation through complex scenes. However, while these representations have many exciting applications, the concern for invasion of individual privacy is significant. Without third parties that we can trust, it is increasingly important to develop scene representations that safeguard individual identity and other sensitive information contained inside visual data.

In this paper, we introduce a framework that connects multiple camera views together without revealing private information about *any* of the camera views (Figure 1 right). Our approach is based on the observation that any Boolean circuited can be cryptographically distributed between multiple parties such that each party can only learn the output of the circuit and nothing else [38, 39]. Our main result is that neural network inference for navigation is compatible with secure multiparty computation and can be distributed between multiple camera views in this way. Our model creates a

36th Conference on Neural Information Processing Systems (NeurIPS 2022).

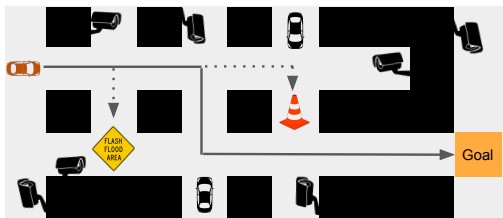 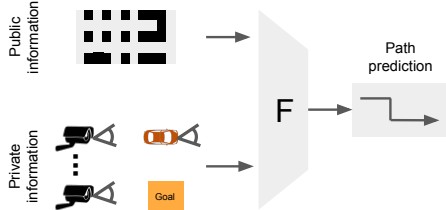

Figure 1: Secure multiparty computation for perception and navigation through cluttered environments. (**left**) We show a navigation task where the red car needs to navigate to the goal position, but there are obstacles that are occluded. Calculating the most efficient path to the destination requires planning ahead to avoid the obstacles. (**right**) Our framework creates a representation from multiple camera views that preserves privacy by only allowing the representation to be used for path prediction, and nothing else (such as facial recognition).

representation of the scene that prevents any other party from accessing the contents of the scene, and guarantees that the representation can *only* be used for predicting the actions needed for the navigation task (and not face recognition, for example).

On a new navigation dataset that we will publicly release, experiments show that private multiparty representations allows navigation through complex scenes while preserving privacy. Compared to plaintext models, our encrypted model results in only a minimal drop in navigation success rate (a gap less than 1%). The approach scales to an arbitrary number of camera views while preserving privacy, making it possible to scale the method to large spaces. When there are multiple valid paths from the source to the destination, the method frequently finds the most efficient path.

The main contribution of this paper is a framework for multiparty perception, which preserves privacy for navigation tasks. The remainder of this paper will describe the architecture and representation for performing this task. Section 2 briefly reviews the related work in multiparty computation and robotic path planning. Section 3 presents our approach to securely distribute the computation among multiple parties, where each party is a camera. Section 4 analyzes the performance and efficiency of our approach on a new navigation dataset. Due to the ubiquity of cameras today, we believe developing visual representations that preserve privacy is increasingly important for many applications such as navigation, and we will publicly release all code, data, and models. We call our method CipherNav.

## 2 Related Work

We briefly review related work in multiparty computation, visual navigation, and robot path planning. This paper integrates secure multiparty computation into visual navigation.

**Multi-party computation (MPC).** Secure multi-party computation (MPC) [38] is a major subfield of cryptography that allows mutually distrusted private data owners (i.e. multiple parties) to compute a function over their private inputs with minimal knowledge transfer [27, 8, 39]. The proposal of MPC has led to many interesting applications, such as Yao's millionaire's problem where two millionaires try to figure out who is richer without revealing their actual wealth [38, 18, 11]. Other applications include secure auctions [3], secure machine learning [16, 17, 30], privacy preserving genomics[35, 12], and more [8]. A common example of MPC protocol includes secret sharing where private data owners split their secrets in into $n$ shares [27, 8]. Knowledge of t shares are able to reconstruct the secret whereas knowledge of $t - 1$ shares are not able to reconstruct the secrets [1, 8, 2, 27]. The majority of the discussion in this paper assumes $t = n$.

**Multiparty computation in machine learning.** Privacy preserving machine learning has gathered much interest in recent years to integrate multi-party computation with machine learning [16, 30, 25, 17, 34, 21, 6, 33]. There are protocols that assume two-party [25, 21], three parties [17, 30, 33], four parties [6], and arbitrary parties [16] computation. Most frameworks assume semi-honest security [16, 30, 25, 21] while others are secure against malicious parties with honest majority assumption [34, 6]. Security against malicious parties comes with computation overhead [17]. However, any known protocols with semi-honest security can be converted to malicious security with an additional zero knowledge proof [19]. Frameworks in this paper assumes semi-honest security level.

**Partial observability in robotic path planning**. Partial observability and the lack of full knowledge of world state have been a long-standing problem in robotic navigation [23, 28, 15]. The existing

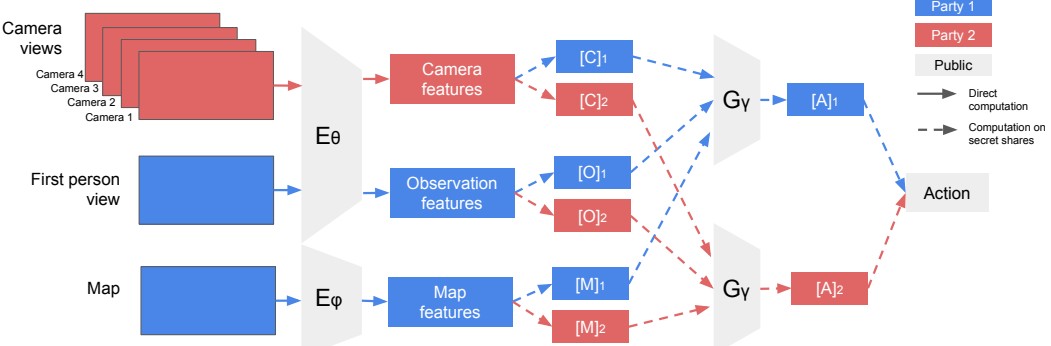

Figure 2: CipherNav action prediction network architecture (for clarity, we only illustrate the two-party case). Public information is gray, agent's private information is blue and external cameras' private information in red. Private operations are locally computed by each party and secret shared in an encrypted manner.

solution either assumes fully observable and predictable environments [9, 31, 37, 20], or attempt environment mapping to achieve better navigation [36, 22]. Existing solutions include modelling navigation as partially observable markove decision process (POMDP) [14, 29] or reconstructing the scene using neural fields to reason about occluded regions through attention based networks [32].

## 3 Multiparty Path Prediction

We present a model to utilize additional private information to achieve better navigation path planning in a privacy preserving way. Particularly, in self-driving navigation and robotic path planning contexts, secure multi-party computation and secret sharing protocols can be used to encrypt the contents from each camera such that the representation can only predict navigation steps. By definition, no parties will be able to infer additional information other than what is reasonably inferred from the final action prediction sequence output [8, 1, 38, 39].

### 3.1 Action Prediction for Visual Navigation

Our goal is to predict the sequence of actions $\{a_0, ...a_t\}$ needed to reach a goal destination in an environment, where the discrete action space $a \in \mathbb{R}^5$ consists of the four directions along a compass (move north, south, east, or west) in addition to an action that corresponds to remaining still. In order to produce this action sequence, the model will be conditioned on the agent's egocentric view $o$ and a map of the environment $m$, which contains public information about the layout of roads. The maps can also contain information that is private to themselves, such as their goal destination. However, the maps will lack any information that is private to other agents, such as the locations of objects and obstacles. The obstacles could block a path, requiring the agent to take a detour.

In order to produce paths that efficiently navigate around obstacles, including obstacles occluded to the agent, our model will additionally condition on a set of multiview camera images $\{c_i\}_{i=0}^n$, where each camera $i$ will produce an image. Multiview images make the environment fully observerable (albeit encrypted in our case). We write our model as $F$, which predicts the next action to perform $a$ given the observations:

$$\hat{a}_{\text{next}} = F(\{c_i\}_{i=0}^n, o, m) \quad \text{s.t.} \quad m_{\text{next}} \leftarrow \Omega(m, a) \quad \text{and} \quad o_{\text{next}} \leftarrow \Omega(o, a) \tag{1}$$

where $F$ is a neural network that we will describe next. After each step, the agent is able to receive the updated map and the updated egocentric view. We denote these updates from the world with $\Omega$.

### 3.2 Encryption and Secret Sharing

Unlike conventional multiview representations, we need to ensure that no information about each party is shared to other parties, except for the information necessary for navigating to the goal destination. We will achieve this by creating representations for each observation that is encrypted. Let $E_\theta(c_i)$ be the feature encoder for the camera and agent views, and let $E_\phi(m)$ be the feature

encoder for the map. Our model learns to predict the action with privacy guarantees through the arithmetic decomposition:

$$F(\{c_i\}_{i=0}^n, o, m) = \sum_{p=0}^{P} G_\gamma \left( [C]_p, [O]_p, [M]_p \right) \mod Q \tag{2}$$

which operates over a finite field $Q$. The summation is over all $P$ parties in the system. We use the notation $[C]_p$ to denote an encrypted representation of the scene, which is the secret share that party $p$ receives for all camera views. It is computed through the concatenation:

$$[C]_p = \left[ [E_\theta(c_0)]_p, \ldots, [E_\theta(c_N)]_p \right] \quad \text{s.t.} \quad E_\theta(c_i) = \sum_{p=0}^{P} [E_\theta(c_i)]_p \mod Q \tag{3}$$

where the constraint on the right hand side follows the arithmetic secret sharing protocol. Each $E_\theta(c_i)$ can be broken into $P$ secrets, denoted as $[E_\theta(c_i)]_p$, which are only sent to their respective party. Each single $[E_\theta(c_i)]_p$ is constructed such that it is insufficient to reconstruct anything about the scene other than what is deducible from the final outputs [27, 8]. However, operations can still be performed on $[E_\theta(c_i)]_p$ such that when all shares are combined, the training task (and only the training task) can be predicted. We similarly define encrypted representations for both the agent's own view and the map:

$$[O]_p = [E_\theta(o)]_p \quad \text{s.t.} \quad E_\theta(o) = \sum_{p=0}^{P} [E_\theta(o)]_p \mod Q \tag{4}$$

$$[M]_p = [E_\phi(m)]_p \quad \text{s.t.} \quad E_\phi(m) = \sum_{p=0}^{P} [E_\phi(m)]_p \mod Q \tag{5}$$

Crucially, this establishes a private visual representation where each party receives just shares of the secret $[C]_p, [O]_p, [M]_p$, which corresponds to a random large integer that does not reveal any information about the original secret feature.

This formulation extends to an arbitrary number of parties. In our experiments, we demonstrate results for both $P = 2$ parties and $P = 5$ parties. As long as one party is honest in this protocol, the full protocol remains secure [1]. Assuming one piece of private information per party, with an increased number of parties, more information is given to the action predictor network.

### 3.3 Learning and Optimization

We use the mean-squared loss function to train the model $F$ in ciphertext. For training, we assume we have a collection of ground truth sequences for the optimal path between a source and destination. We optimize the following learning problem with stochastic gradient descent:

$$\min_{\theta, \phi, \gamma} \mathbb{E}_{(c, o, m, a)} \left[ (\hat{a} - a)^2 \right] \quad \text{where} \quad \hat{a} = F(\{c_i\}_{i=0}^n, o, m) \tag{6}$$

Since action sequences are variable length (up to a maximum), the model also predicts a stop symbol represented by 0. We leverage the stop token to mask the section of the sequence following the stop token to prevent gradient back propagation in steps after the stop token. We use a one-hot vector to represent the action space $a \in \mathbb{R}^5$, where the $\arg\max$ corresponds to the action to perform.

We originally experimented with a cross-entropy loss, but the optimization failed to converge. Computing cross entropy requires logarithmic and exponential functions that are hard to compute in multi-party computation settings. For example, logarithmics are evaluated through Householder iterations [10] and exponentials need to use limit approximation [16]. The approximations affect performance and lead to numerical overflow issues in practice.

We split our encryped training scheme into two stages. First, since the feature extraction does not require secret sharing, the view encoder $E_\theta$ and the map encoder $E_\phi$ are pretrained in plaintext. We compute the camera features, observation features and map features locally in order to save computation cost, as the multi-party computation is expensive. In the second stage, where secret sharing is paramount, we replace the G action classification network with a new encrypted G that we train with multi-party computation from scratch, while freezing the pre-trained encoders.

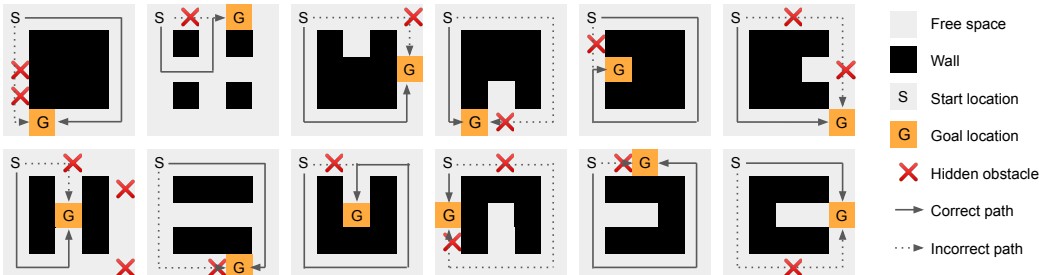

Figure 3: Top view visualization of random samples from the Obstacle World dataset. There are 12 random permutations of the map states. Each map has hidden obstacles that will impede the agent from reaching the goal. The agent may need to make detours, but least one viable path exists.

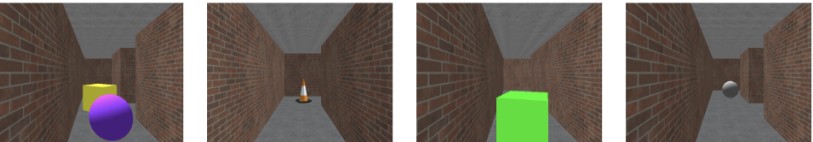

Figure 4: Possible obstacles visualization as seen by multi-view cameras. Obstacles occur in random locations, colors, and shapes.

### 3.4 Implementation Details

The network encoders include the view encoder $E_\theta$ and map encoder $E_\phi$. The view encoder $E_\theta$ consists of two layers of convolutional layers followed by two linear layers, with ReLU activations in between. $E_\theta$ takes in a 45 x 60 dimensional input view image and transform it into 32-dimensional vector. The map encoder $E_\phi$ consists of 3 linear layers with ReLU activations. The map encoder takes in a discrete square map represented by 5x5 matrix, and outputs map features as a 128-dimensional vector. The view image features and map features are passed into action classification network $G$, which is another multi-layer perceptron with four linear layers and ReLU activations. Only ReLU is chosen as the activation function due to the increased complexity involved to approximate other functions in ciphertext.

We implement CipherNav in PyTorch [24] and use the Crypten [16] framework for privacy-preserving neural network operations. Each model is trained with 600 epochs and 0.01 learning rate. The batch size equals to 500 for ciphertext models and 100 for plaintext models. The plaintext models are trained on a single GPU for 1.5 days. The ciphertext 2-party computation models are trained on a single GPU for 3 days.

## 4 Experiments

### 4.1 Obstacle World Dataset

To analyze our framework, we construct a new dataset, The Obstacle World, which we generated based on the open-source Gym MiniWorld environment [7]. Each instance of the environment is randomly generated out of 12 possible permutations of the map as shown in Figure 3. The agent starts at the top left corner of the map, and chooses a randomized goal location. However, obstacles are scattered throughout the world that may prevent the agent from reaching the goal in shortest path. There are 1 to 3 obstacles generated with random locations, random shape, and random colors 4. The cones are rare wild card obstacles with lower chance of appearance compared to other obstacles. The dataset assumes that there is at least one possible obstacle-free path from the agents' starting location to its' goal location, though multiple paths often exist. The training and validation datasets are also engineered to be balanced. In 50% of the cases, obstacles lie on the shortest path and the agent has to make a detour to reach the goal. In the other 50% cases, no obstacles lie on the shortest path and a detour to reach to goal will make the navigation planning less efficient. We use $15,000$ environments for the training set, and $2,250$ environments for the testing set, which is disjoint.

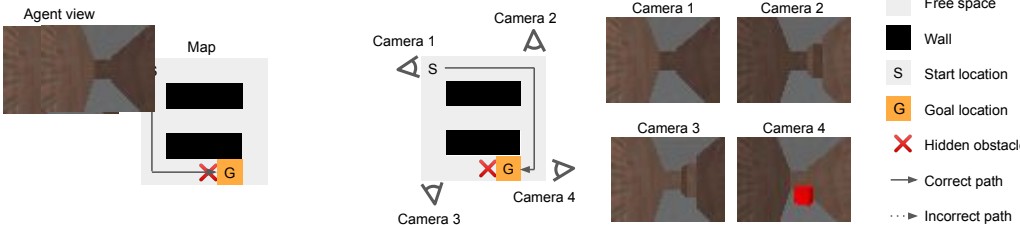

Figure 5: Without multiple views, the agent fails to avoid obstacles en route to the goal.

Figure 6: With multi-view observations from many cameras, the agent reaches the goal while avoiding obstacles. The red cube obstacle in camera 4 corresponds to the red cross in the map.

The dataset is challenging to solve when the agent only has partial observability and imperfect knowledge of the world state. Each agent is given a map, which includes public information such as roads and walls as well as private information such as current start location, end goal location, and first-person view. The agent tries to reach the goal with the shortest path. However, as seen in Figure 5, failure cases often occur when agents fail to take into consideration obstacles beyond their own first person view observations.

The presence of multi-view observations (which our method encrypts) offers additional information that help agents reason about hidden obstacles in Figure 6. There are four fixed location cameras in Obstacle World to provide multi-view observations. We set a camera at each of the four corners, such that each camera is able to look over an entire lane. Additionally, there exists a fifth camera, which is the agent's first person point of view. At each time step, a new image is generated from the agent's view, which changes as the agent takes steps forward. The four cameras offer full observability to the world state, allowing our network to estimate about whether a detour is necessary.

## 4.2 Baselines and Methods

We compare against multiple methods that predict action sequences. These methods have varying privacy guarantees and input information.

**Random walk**. In the random walk method, the agent has no access to any additional information in the environment besides the starting location, and has to make random guesses to possibly reach towards the goal state. The agent is able to freely pick an action till a maximum number of steps without any knowledge on the possible path reaching goal state. To avoid an almost zero success probability, the random walk baseline is designed to be clash-free, meaning that the agents will only choose available steps that do not result in a clash to the wall or to the obstacles.

**Map only**. We train a network to predict action sequences based exclusively on the starting location, the goal location, as well as the input map. This means the agent is able to access public information such as roads and walls, similar to how we are able to access fixed information from google maps. This approach has a strong privacy guarantee because all operations are performed locally by the agent. A multi-layer perceptron is used as a map encoder, and the output features are fed into another multi-layer perceptron to classify the action.

**First person view**. This is strongest achievable baseline without any privacy compromise, where the agent has access to the starting location, the goal location, the map, and its' own first person point of view. No multi-party computation is involved as all the private data belongs to the agent and local computation is possible. There are two difficulty levels for the agent. The deterministic start baseline assumes that the agent is always entering from left to right and has full view of the top lane. Hence, the agent is able to effectively learn that obstacle in view means do not enter the top lane. The random start baseline assumes that agent can enter from left to right or top to bottom with 50% probability each. Hence, in the random start baseline, the agent view does not help in taking the first step.

**Encrypted camera views (our method)**. We propose to train the network with the same inputs as plaintext with camera views above, i.e. the starting location, the goal location, the agent's own first-person view, as well as the multi-view observations, all via a neural network with multi-party computation. The network architecture is seen in Figure 2. Each piece of private information is broken down into 2 or 5 shares depending on the number of parties involved. One of the parties is

Table 1: Plaintext and ciphertext test accuracy. The last experiment plaintext with camera views allows full information access while breaching privacy. All other approaches provide strict privacy guarantees. All start views are random unless otherwise specified as deterministic start.

| Experiment | Detour Required | No Detour | Overall |
|---|---|---|---|
| Random | 6.2% | 53.1% | 29.9% |
| Map only | 64.3% | 77.5% | 74.0% |
| First Person | 64.5% | 79.7% | 72.0% |
| First Person (deterministic starting view) | 86.0% | 91.6% | 88.8% |
| 2-party MPC w/ Camera Views | 93.6% | **98.8%** | **96.6%** |
| 5-party MPC w/ Camera Views | **94.3%** | **99.2%** | **96.9%** |
| Plaintext w/ Camera Views | **95.2%** | **99.5%** | **97.1%** |

always the agent taking an action. In the 2-party setting, all the security cameras are owned by one party, such as the government. This assumes the security cameras trust each other. In the n-party setting, each security camera is owned by a different party, so there are many parties. This assumes the security cameras do not trust each other. The CipherNav models are both trained and evaluated in privacy-preserving ways through secret sharing protocols. The experiments assume public network parameters and private input data. Multi-party computation assumes that if at least one party is honest, then the protocol preserves privacy. Hence, any agent who wants to keep its information private will have strong incentive to remain honest, rendering the entire privacy guarantee secure.

**Plaintext with camera views**. To understand the performance in the idealized case, we train a network similar to our method, but without any privacy guarantees. This neural network uses an architectures similar to Figure 2, except that everything is trained in plaintext without multi-party computation. As such, either the agent has full knowledge of the security camera images, or the third-party has full access to the agent's first person camera view sequence, goal location, and current location. Plaintext training assumes the existence of a trusted third party with full knowledge of all the private information.

## 4.3 Quantitative analysis

Table 1 shows that our proposed MPC models achieve significant improvement over other privacy-preserving baselines. Furthermore, our privacy-preserving model's results are comparable to the accuracy of the non-privacy preserving models (less than 1% difference in performance).

To provide quantitative analysis on the effectiveness of CipherNav, seven experiments are conducted with five plaintext baselines and two ciphertext multi-party computation models. With maps indicating the roads and walls, the agents are only able to get 64.3% accuracy in detour required cases and 77.5% accuracy in no detour cases. This is significantly lower than the plaintext results with multi-view camera supervision, where the accuracy is 95.2% in detour required cases and 99.5% in no detour cases. However, the plaintext with camera views experiment assumes full knowledge of camera images and provides no privacy protection to agents.

Other privacy-preserving baselines are also calculated in cases where the agents' first person view is provided. The "first person" model barely improves in accuracy as compared to the map only model, assuming the default random start. In other words, due to the number of occluded obstacles in our dataset, a first person view is not helpful when a map and agent location is provided.

We additionally compare the "first person" model with a random starting view direction to another model trained with a deterministic starting view direction. We notice a significant improvement in accuracy, and attribute this to the fact that a deterministic starting view would allow the model to orient the agent with respect to the map at the very beginning. In other words, the agents always have full information on the top lane to better choose the first step, and hence the accuracy increases to 86% for detour required case and 91.6% for no detour case. However, the overall accuracy is still 8.3% below the theoretical best result assuming plaintext multi-view camera observation.

In our proposed models, 2-party MPC with camera views and 5-party MPC with camera views both achieve signfcant improvement with 96.6% and 96.9% overall accuracies respectively. In the 2-party MPC experiment, we find that all three experiments of "Detour Required," "No Detour' and "Overall" experience a less than 2% drop in accuracy as compared to the upper-bound plaintext accuracy,

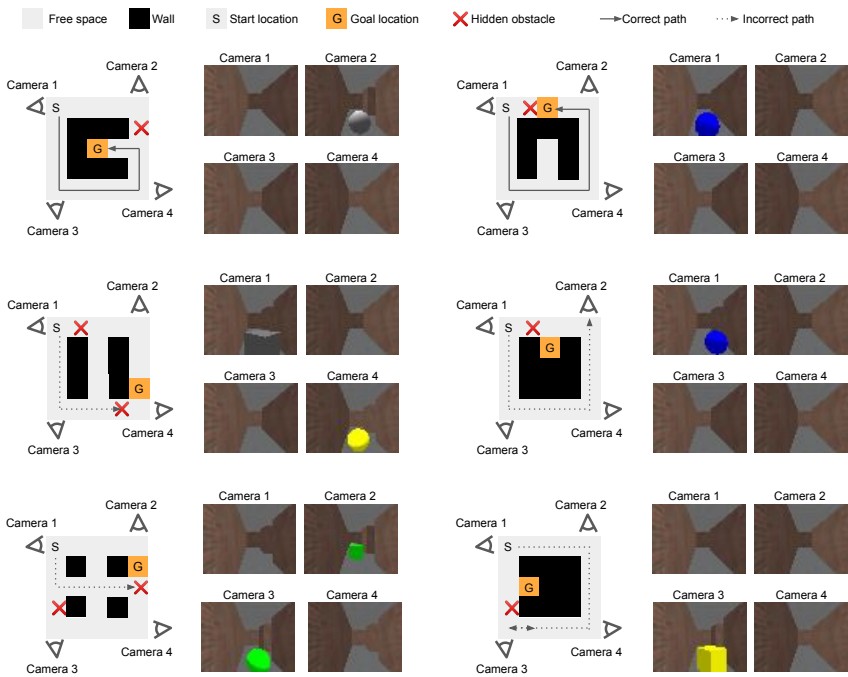

Figure 7: We visualize several examples of our agent navigating The Obstacle World while simultaneously preserving privacy. The first row shows success cases, and bottom rows show instructive failures.

while providing full privacy guarantees. Similarly, in the 5-party MPC experiment, we find all three experiments have a less than 1% drop in accuracy as compare to the upper-bound plaintext accuracy, while providing full privacy guarantees. This highlights the applicability of multi-party computation in private perception navigation tasks.

## 4.4 Qualitative Analysis

Qualitative analysis results are shown in Figure 7 to illustrate success and failure cases provided by model outputs. The top row illustrate the two success cases. With multi-view supervision, the agent is able to perceive hidden obstacles and navigate complex detours to reach goal locations, even if the goal location is deep in to a dead end. The second row illustrates the cases where the agent fails due to overly long and complicated scenarios. Four or five turns are required to maneuver the obstacles, which proven to be a challenge for both plaintext and ciphertext models. Solving the case requires more complex network architecture.

The two examples in the last row highlights the possible drawbacks of the ciphertext model after taking in additional information. In the first case, the agent could easily reach the goal by following the shortest path, but instead chooses to make an unnecessary detour. In the last example, the agent is stuck in an indecisive state, where the observations from camera views to avoid obstacles come into conflict with the understandings from maps to reach the goal. Upon reaching the bottom left corner, the observations from the multi-view cameras indicate the presence of an obstacle, prompting the agent to move one step back towards the right. However, since decisions are made agnostic about previous states, the agent repeats the same mistake and step forward in attempt to reach the goal. While additional private information hugely enhances the potential navigation outcomes, these examples illustrate the needs to account for edge cases and potential information conflict after including additional private information in downstream tasks, even if privacy is no longer a concern in the proposed models.

## 4.5 Inference Runtime and Result Distribution Analysis

To quantify the inference runtime on multi-party computation, Table 2 records the forward pass time on both GPU and CPU on input data with batch size 100 on the action prediction network only. Features are pre-computed for both plaintext and ciphertext networks. Multi-party computation is

Table 2: We show inference time comparison between ciphertext and plaintext models.

| Experiment | Runtime (seconds) | |
| | GPU | CPU |
| --- | --- | --- |
| plaintext | 0.00028 | 0.00032 |
| 2-party | 0.15 | 0.23 |
| 5-party | 0.42 | 1.3 |

Table 3: Failure counts across different models out of 4500 trials

| Experiment | Crash Obstacle | Crash Wall | No Crash |
| --- | --- | --- | --- |
| Map only | 1309 | 0 | 0 |
| First Person | 1252 | 0 | 0 |
| First Person (det. start view) | 491 | 13 | 0 |
| 2-party MPC w/ Camera Views | 144 | 16 | 10 |
| 5-party MPC w/ Camera Views | 115 | 21 | 11 |
| Plaintext w/ Camera Views | 105 | 2 | 11 |

more expensive where forward pass in two-party computation has a 500x overhead on GPU and forward pass in five-party computation has a 1500x overhead on GPU. The large computational overhead highlights the need to perform computation in stages and off-loading computations to non-MPC stages.

We experimented with the success and failure counts for different models over different path lengths. While all four models achieve similar success count in shorter path length $\leq 4$, the difference in success count becomes obvious after path length $> 5$. On path length $\geq 9$ where detours are more necessarily to reach the goal, we found plaintext camera views and multi-party computation models perform signficantly better with lower failure rate and higher success rate.

Table 3 breaks down reasons of failures into three categories: crash obstacle, crash wall, or no crash. No crash failure case usually refers to the scenario where the agent is stuck in an indecisive state due to conflicting information from external camera views and internal map and first person views. A simple map baseline model can effectively learn not to crash into the walls, but has a significantly higher rate of crashing into hidden obstacles. The introduction of additional information brings additional challenges but significantly reduce the chance of overall hazards. However, the majority of the limitations remain in plaintext, and the introduction of encryption and multi-party computation does not seem to result in major decrease in performance.

## 5   Conclusion

Navigation is a fundamental challenge in computer vision and robotics, and the task is difficult for partially observerable environments that have many occlusions and visual obstructions. We have demonstrated a framework that integrates secure multiparty computation with multiview vision in order to create a private representation for navigation. Due to the number of applications for navigation and the importance of privacy, we believe integrating the two areas is a promising direction to deliver robust yet secure navigation policies for robotics.

**Limitations and Future Work.** This paper is only the first step, and there remains many important steps in order to realize the tangible applications of this method. Our method is designed for both fixed cameras positions and static environments, and future work will need to expand the architecture in order to relax this assumption. Additionally, an important limitation towards real-world deployments is the wide area network latency and bandwidth, which must be sufficiently fast to transfer shares between parties. Overcoming this challenge will require interdisciplinary advances at the intersection of computer networking, cryptography, and machine learning. Finally, multi-party computation introduces additional overhead to inference, motivating the need for a new generation of neural network architectures that are designed to be efficient under privacy guarantees.

**Societal Impact.** Our research is founded on ethical considerations, and we are excited for the potential for navigation to push the frontiers in robotics and assistive technology, such as navigation for people with disabilities. Due to the ubiquity of cameras today, we believe visual representations that tightly integrate with cryptography and privacy-preserving techniques will be critical for delivering the future applications of computer vision, and we hope this paper spurs additional work along this direction. If correctly implemented, multiparty computation provides rigorous guarantees that a party cannot learn anything more from the computation than the task itself.

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
