# 6 Supplementary Material

## 6.1 Network Architecture

The section explains detailed CipherNav network architecture in Table 4, 5 and 6. The view encoder $E_\theta$ is shown in Table 4 and map encoder $E_\psi$ is shown in Table 5. The encoders are trained end-to-end during plaintext training and freezed during ciphertext training. Each party has a copy of the encoder models and locally computes all forward passes in ciphertext training. The action classification network $G$ is shown in Table 6. The network $G$ is assumed to have public parameters, but the input values are private. Secret sharing protocols are used to jointly compute an action prediction sequence without revealing any additional information on the private inputs.

Table 4: CipherNav view encoder detailed network structure. Layer type C indicates Conv2D layer and layer type L indicates Linear layer. A ReLU activation function is added in between each layer.

| | View Encoder | | | |
|---|---|---|---|---|
| Number | 1 | 2 | 3 | 4 |
| Type | C | C | L | L |
| Input Size | 3x45x60 | 6x21x28 | 648 | 128 |
| Output Size | 6x21x28 | 6x9x12 | 128 | 32 |
| Kernel | 5 | 5 | - | - |
| Stride | 2 | 2 | - | - |
| Params | 456 | 906 | 83072 | 4128 |

Table 5: CipherNav map encoder detailed network structure. Layer type L indicates Linear layer. A ReLU activation function is added in between each layer.

| | Map Encoder | | |
|---|---|---|---|
| Number | 1 | 2 | 3 |
| Type | L | L | L |
| Input Size | 25 | 64 | 128 |
| Output Size | 64 | 128 | 128 |
| Kernel | - | - | - |
| Stride | - | - | - |
| Params | 1664 | 8320 | 16512 |

Table 6: CipherNav detailed network structure for the action classification network. Layer type L indicates Linear layer. A ReLU activation function is added in between each layer.

| | Action Classification Network | | | |
|---|---|---|---|---|
| Number | 1 | 2 | 3 | 4 |
| Type | L | L | L | L |
| Input Size | 288 | 128 | 64 | 16 |
| Output Size | 128 | 64 | 16 | 5 |
| Params | 36992 | 8256 | 1040 | 85 |

## 6.2 Dataset Generation

The Obstacle World training and testing datasets are generated disjointly. In training dataset, the ground truth actions and shortest paths are pre-generated based on breath-first-search (BFS). 1-3 obstacles are generated across the borders of the map. The dataset assumes that at least one camera is able to see the obstacle. Then, the goal location is generated in one of the reachable locations given the current location and obstacle locations.

During training, the network assumes teacher forcing. Regardless of the predicted training output, the network will assume the ground truth label to update the map and environment. The map is represented by a 5x5 matrix where each number has a separate meaning. Given an action, the current location on the map can be updated to generate a new map. To update the environment, the agent moves through the maze, and its' first person view is recorded per time step.

The testing dataset is generated on the fly. A new environment is generated each time where the agent moves through the Obstacle World based on actions given by the model prediction. The first person views and maps are updated according to model outputs. As such, a new testing dataset of size 2250 is randomly generated each time a new model is tested. The distribution of the test datasets remain the same across different baselines.

## 6.3 Arithmetic and binary secret sharing

There are two types of secret sharing protocols: arithmetic and binary protocols. The arithmetic secret sharing protocol requires finite field and all operations are performed through addition and multiplication. All other operations such as division and non-linear function involves approximation to convert them into addition and multiplication based operations [16]. Note that all operations below are modular over finite field with $Q$ elements.

**Arithmetic addition** Assume $x, y$ indicate secrets, $[x], [y]$ indicate arithmetic shares of secrets and p indicates party, to calculate addition $z = x + y$, we first break down x and y into shares $x = \sum_{p \in P} [x]_p \mod Q$ and

$y = \sum_{p \in P} [y]_p \mod Q$. Shares of $z$ can be calculated through $[z]_p = [x]_p + [y]_p \mod Q$ and $z = \sum_{p \in P} [z]_p \mod Q$.

**Arithmetic Multiplication** Multiplication is more complicated because each multiplication requires Beaver's multiplication triplet $([a], [b], [c])$ where $c = ab$. The triplet can either be pre-computed by trusted third parties or securely generated on the fly through oblivious transfer. During multiplication computation $z = xy$, each party has a share of $x, y, a, b, c$, but not the actual values. Each party performs computation on $[\epsilon]_p = [x]_p - [a]_p$ and $[\sigma]_p = [y]_p - [b]_p$, and then broadcast the share of $[\epsilon]_p, [\sigma]_p$ in a communication round. All $P$ shares of $[\epsilon], [\sigma]$ are used to reconstruct $\epsilon, \sigma$. All parties are aware of the value of $\epsilon = x - a$ and $\sigma = y - b$, but the remain agnostic to the values of $x, y, a, b, c$. To compute share of $[z]_p$, each party calculates $[z]_p = [c]_p + \epsilon[b]_p + [a]_p\sigma + \epsilon\sigma \mod Q$. A simple arithmetic calculation reveals that $z = c + \epsilon b + a\sigma + \epsilon\sigma = ab + (x-a)b + (y-b)a + (x-a)(y-b) = xy$. The final product $z = \sum_{p \in P}[z]_p \mod Q$. Since the last term $\epsilon\sigma$ is calculated in plaintext, only one party needs to add $\epsilon\sigma$ when computing the share.

**Binary XOR** Bitwise XOR operation is similar to the addition operation in arithmetic secret sharing. Let $\langle\rangle$ denotes binary secret sharing operations, $\langle z \rangle_p = \langle x \rangle_p \oplus \langle y \rangle_p$.

**Binary AND** Bitwise AND operation is similar to the multiplication operation in arithmetic secret sharing. AND operation require pre-generated Beaver's triplet $(\langle a \rangle, \langle b \rangle, \langle c \rangle)$ where $c = a \otimes b$. The parties similarly compute $\langle \epsilon \rangle_p = \langle x \rangle_p \oplus \langle a \rangle_p$ and $\langle \sigma \rangle_p = \langle y \rangle_p \oplus \langle b \rangle_p$, and publicly broadcast their shares of $\epsilon$ and $\sigma$. $\langle z \rangle_p = \langle x \otimes y \rangle_p = \langle c \rangle_p \oplus (\epsilon \otimes \langle b \rangle_p) \oplus (\langle a \rangle_p \otimes \sigma) \oplus (\epsilon \otimes \sigma)$.

## 6.4 Path planning efficiency analysis

Our proposed CipherNav has achieved the highest efficiency (or within 1% difference in percentage) across no detour, detour required, and overall categories in Table 7.

Figure 8 visualizes several cases where the agents fail to reach the goal in the most efficiency path. Such cases occur with very low probability (less than 1%), and the reason of occurrence is often arbitrary. Comparing privacy preserving MPC models with non-privacy-preserving plaintext models in Table 7, the highest and lowest efficiency values are less than 1% apart. In detour required case, the efficiency is close to 100% because the correct path is a relatively longer path by definition. Hence, given the agent successfully reaches the goal, it is highly unlikely that the agent chooses an even longer path. In the no detour category, the shortest path equals to the correct path. Hence, it is more likely that multiple paths with longer path length exist, leading to slightly lower efficiency across all baselines. Our proposed CipherNav model is approximately 3-4% more efficient compared to the map only and first person baselines in the overall category, and 6-7% more efficient in the no detour category.

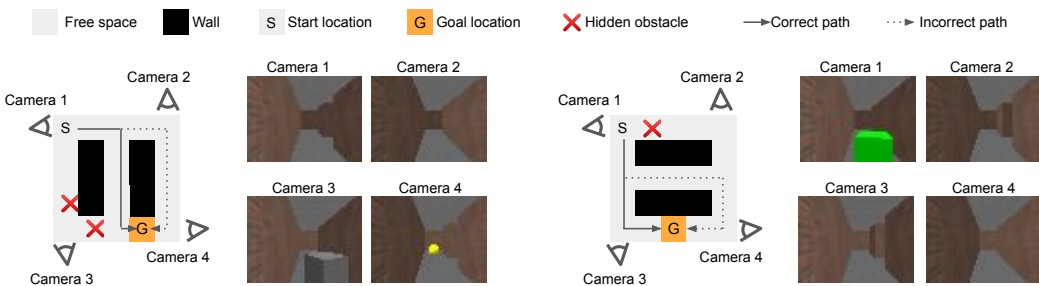

Figure 8: We visualize several examples of our agent fail to reach the goal in the most efficient path.

Table 7: Efficiency of path found in plaintext and ciphertext models. The efficiency denotes percentage where the agents reach the goal in shortest path given that the agents successfully reach the goal.

| | Efficiency | | |
| --- | --- | --- | --- |
| Experiment | No Detour | Detour Required | Overall |
| Random | 38.8% | 7.2% | 37.8 % |
| Map only | 93.2% | **100%** | 95.5 % |
| First Person | 94.1% | **100%** | 96.8 % |
| First Person (deterministic starting view) | 93.7% | **99.9%** | 96.6 % |
| 2-party MPC w/ Camera Views | **99.1%** | 99.1% | **99.5 %** |
| 5-party MPC w/ Camera Views | **100%** | 99.5% | **99.2 %** |
| Plaintext w/ Camera Views | **99.5%** | **99.7%** | **99.4 %** |

## 6.5 Privacy guarantees and limitations

Theoretical results have been rigorously established that prove multi-party computation (MPC) allows multiple parties to jointly compute over private inputs without revealing any information other than what's reasonably deductible from the outputs themselves [27]. Our approach leverages this result and applies it to the navigation problem.

Our proposed framework provides a full privacy guarantee against any third-party attacks, provided that the parties do not voluntarily reveal their secret shares themselves. For a third party to attack the system, all parties need to be dishonest [27, 8, 1]. The benefit of our system with MPC is that the secret owner is one of the n parties, who have full incentive to hide their secret shares to prevent information leaks. The incentive alignment allows our framework to satisfy all-but-one honest security requirement of MPC. Semi-honest security level assumes that all parties are curious about sensitive information but not malicious. In other words, while full privacy is guaranteed nevertheless, the agents and the security camera owners like the government are trusted to follow the MPC computation protocol for the final navigation results to be accurate. Following the MPC protocol is aligned with the agents' incentives to have a better navigation experience and the government's incentives to build a better smart city. To provide rigorous guarantees of output accuracy, zero-knowledge proofs can be added as follow-up work.

In multi-party computation, the network outputs naturally reveal some limited information on the inputs. One possible privacy attack is for a user to repeatedly query the network multiple times. If the user were to perform such an attack, the most they could infer is the location of an obstacle, but nothing other than the existence of that obstacle (i.e. identities of the obstacle, category of the obstacle, or other scene features). By querying the network to exhaust all goal locations, the user can obtain a set of possible trajectories. By looking at the regions the trajectories never go to, the user could infer the location of possible obstacles. Consequently, the upper bound on the revealed information is the potential obstacle locations but nothing else.

Table 8: A scene reconstruction network is trained to predict the existence of the obstacle in a camera view. The experimental results are in good agreement with the theoretical results, where the prediction given encrypted features is no different from a random guess.

|  | Accuracy |
| --- | --- |
| Random chance | 50% |
| Encrypted ciphertext feature (ours) | 49.7% |
| Non-encrypted plaintext feature | 93.6% |

Although privacy is guaranteed by theoretical results, an additional experiment is conducted to show that experimental results are in good alignment with the theoretical guarantees. In Table 8, a scene reconstruction experiment is conducted to reveal the existence of obstacles in a camera view given the feature vector extracted by the camera. The model used is a linear classifier with cross-entropy loss, although other models will yield the same results. All features are normalized to the range -1 to 1 to avoid precsion issues with large integers. From Table 8, the binary prediction based on encrypted ciphertext features has close to 50% accuracy, which is indistinguishable from a random distribution.