# OpenReview forum: "Private Multiparty Perception for Navigation"
_NeurIPS.cc/2022/Conference — NeurIPS 2022 Accept_

### Official Review · Reviewer_N4st · 2022-07-11

**Rating:** 4
**Confidence:** 2
**Soundness:** 3 good
**Presentation:** 3 good
**Contribution:** 2 fair

**Summary:**

The paper presents CipherNav, a novel framework for private multiparty perception for navigation. CipherNav aims to predict the sequence of actions needed for an agent to reach a goal destination. The inputs to CipherNav are (a) camera views from P participants, (b) the agent's ego-centric camera view, and (c) a map of the environment containing public information about the layout of roads. In order to preserve privacy between the participants, inputs (a), (b), and (c) are encrypted via arithmetic decomposition, such that each party receives a partial share of the secrets. The predictions predictions from each participant are then arithmetically combined to produce the final action sequence. In order to validate CipherNav,  a novel dataset based on the open-source Gym MiniWorld environment is generated with 5x5 maps. Obstacles are randomly inserted into the maps to impede the agent from reaching the goal state. Experiments on the generated dataset show that the CipherNav achieves comparable performance to a model operates on plaintext images.

**Questions:**

1. What are the technical contributions beyond applying SMPC to a new problem?

**Limitations:**

The authors adequately addressed the limitations and potential negative societal impact of their work.

**Strengths And Weaknesses:**

**Strengths**
* Novel setting of private mutiparty perception for navigation is introduced. Given the increasing importance of privacy-preserving ML, this is an important setting to consider.
* Novel dataset is introduced. The proposed Obstacle World Dataset, while simple, can serve as a good baseline for the task of private multiparty perception for navigation.
* Results on the novel Obstacle World Dataset show that the proposed approach achieves comparable performance to a model that operates on plaintext images and does so with a reasonable computational overhead.
* The proposed CipherNav network architecture is novel and seems to work well for the proposed task


**Weaknesses**
* Limited technical contributions.

---

> ### Author Response · Authors · 2022-08-02
> **Responses to Reviewer N4st**
>
> Thank you for your comments, and we are glad that you appreciate the novelty of our approach. Our paper makes technical contributions on multiple levels:
>
> New Framework for Navigation: Navigation is a longstanding challenge in the machine learning, robotics, and computer vision fields, with many papers published on the topic every year. However, a robust solution to the problem has remained intractable because environments are cluttered and partially observable. A key technical contribution of our work is showing, for the first time, how to integrate multiple cameras together while simultaneously respecting privacy for navigation tasks. We believe this interdisciplinary result is significant because it introduces a fundamentally different method for navigation, and consequently will receive wide interest.
>
> New Architecture: In order to achieve this, we have introduced a new neural network architecture for navigation that is compatible with privacy-preserving methods and secure multi-party computation. Since it is well known that the performance of neural networks heavily relies on the architecture, we believe this contribution is significant. Secure multi-party computation comes with a different set of constraints compared to the normal neural network setting because a) it introduces fundamental latencies, and b) the network operations need to be aligned with the theoretical guarantees. For example, operations that seem easy in a normal setting, such as exponentials, multiplicative inverse, and non-linear activation, now all come with additional costs, approximations and trade-offs with multi-party computation. Our paper introduces an architecture that supports both properties and judiciously manages the trade-offs without compromising privacy. Moreover, our results show there is very little performance drop when privacy is preserved with our architecture.
>
> We hope this addresses your concerns, and thank you for your consideration.

---

### Official Review · Reviewer_BTzU · 2022-07-12

**Rating:** 6
**Confidence:** 2
**Soundness:** 3 good
**Presentation:** 3 good
**Contribution:** 3 good

**Summary:**

This paper proposes a method for robot navigation from multiple third-person views that additionally attempts to preserve the privacy of the information captured by the cameras. This is accomplished by training the networks used in the model to operate on encrypted feature vectors, thus eliminating the need to share the original data. This allows for navigation without any party having access to all the information. The method is tested on a Gym Mini World map. The results show that the method has comparable accuracy to a non-private baseline.

**Questions:**

- How does the method ensure that each encrypted feature vector can't be used to reconstruct anything about the scene?
- How is the conversion from (assumingly) continuous feature vectors to integers done?
- Does the system need to share keys between the parties?
- Why during training (Eq 6) is finite field arithmetic not used?

**Limitations:**

The paper addresses the societal impacts adequately. However, I would probably suggest including some information on the limitations i.e. perhaps some information about the images could be reconstructed by repeated querying the navigation network? Could the system be attacked by a third party?

**Strengths And Weaknesses:**

Strengths
=========
- Preserving privacy in robotics and computer vision applications is an area of growing importance and thus methods that can help achieve this goal are much needed. This is partially because images usually capture so much additional information that is not particularly needed for a task, and thus sharing them with a third party can have major privacy concerns. This is common in robot vacuum cleaners, for example, where computation is usually limited to the edge in order to preserve privacy. The proposed method is a good solution to this if the processing has to be done remotely.

- The method seems quite novel. While preserving privacy in neural networks is an active area of research, as far as I’m aware there aren’t any learning-based privacy-preserving navigation approaches.

- The method is simple but effective. The method works with standard embeddings computed by a neural network that are then encrypted using modular arithmetic. The network then learns to perform the action using this arithmetic decomposition of the representations. This could be easily applied to any existing navigation solutions that make use of feature representations.

- The results show that the encrypted navigation pipeline is only a fraction of a percent less accurate than the corresponding plaintext pipeline which is quite surprising given that the networks operate on encrypted feature vectors.


Weaknesses
==========
- If the action prediction network can successfully predict actions using the encrypted features, it might be possible to train a similar network to reconstruct the either the multi-view images or the images from the agent. The paper states “Each $\left[E_{\theta}\left(c_{i}\right)\right]_{p}$ is constructed such that it is insufficient to reconstruct anything about the scene” but it’s not clear why this is the case.

- The evaluation is only performed on the Gym Mini World map which is not very realistic. How would the method work in a real-world environment where there is a lot more variation between scenes and the map and cameras exhibit noise, occlusion and artifacts. Would the performance between the plaintext and encrypted methods still be similar or would the proposed approach be affected more by this variation and adverse conditions?

---

> ### Author Response · Authors · 2022-08-02
> **Responses to Reviewer BTzU**
>
> We appreciate your thoughtful feedback and helpful suggestions! There are many great questions asked and we are looking forward to the discussion.
>
> **How does the method ensure that each encrypted feature vector can't be used to reconstruct anything about the scene?**
>
> Thanks for the interesting question. Theoretical results have been rigorously established that prove multi-party computation (MPC) allows multiple parties to jointly compute over private inputs without revealing any additional information other than what’s reasonably deductible from the outputs themselves [27, 7, 2]. Our approach leverages this result, and applies it to the navigation problem. The encrypted feature vectors are shares of the original feature vector. Although the addition of all shares will reconstruct the original feature vector, individual shares are essentially random large integers. The distribution of the encrypted feature vector is indistinguishable from the random feature vectors. You are right that the line 116-117 can be better clarified. We have updated it to “Each [Eθ(ci)]p is constructed such that it is insufficient to reconstruct anything about the scene, other than what’s reasonably deducible from the outputs themselves”.
>
> **Possibility to train a similar network to reconstruct the images**
>
> The encrypted feature vectors are theoretically guaranteed to be indistinguishable from a random signal [27, 7, 2]. To provide an intuition, we also trained a linear classifier to predict if an obstacle exists in the given lane. Based on the non-encrypted feature vector, we can achieve 93.6% accuracy. However, with an encrypted feature vector, the accuracy is 49.7%, which is equivalent to a random guess. Details of the experiment are added to the appendix and our empirical results are highly consistent with the theoretical results.
>
> **Images could be reconstructed by repeatedly querying the navigation network?**
>
> Nothing can be reconstructed other than what could be inferred from the final action prediction outputs [27, 7, 2]. In other words, if the user were to query the navigation network multiple times, the most they could possibly infer is the location of an obstacle, but nothing other than the existence of that obstacle (i.e. identities of the obstacle, category of the obstacle, nor other scene features). By querying the network to exhaust all goal locations, the user is able to obtain a set of possible trajectories. By looking at the regions the trajectories never go, the user could infer the location of possible obstacles. Consequently, the upper bound on the revealed information are the potential obstacle locations but nothing else.
>
> **Could the system be attacked by a third party?**
>
> No, this would not be possible. For a third party to attack the system, all parties need to be dishonest to violate perfect privacy [7, 2, 16, 27]. The benefit of our system with MPC is that one of the parties is the secret owner itself, who has full incentive to remain honest and prevent third parties from stealing the secret. The agents know that their secrets are safe from any other parties as long as they do not reveal their shares themselves.
>
> **Add some information on limitations**
>
> Thank you for the helpful advice. We have added a section 6.5 in the appendix that discusses the limitations.
>
> **Performance on more realistic dataset with more nuances**
>
> Thank you for asking an important question. The scope of the paper is establishing the first framework for navigation with multiparty computation and demonstrating that this is possible. Our paper contributes principles and guidelines of MPC compatible navigation neural architectures that we believe will be adaptable to more realistic datasets. Our paper is the first step to a new direction and we will make all code, models, and datasets available for the community to gradually expand on the complexity of the datasets.
>
> **Would the performance between the plaintext and encrypted methods still be similar under variation and adverse conditions?**
>
> Theoretically, any finite field arithmetic computation in MPC will not incur any accuracy loss, meaning that most standard neural networks layers are theoretically compatible [16, 33, 17]. However, in practice, there are necessary approximations which can create a performance gap. This is an active field of research and our paper starts to bridge this gap for navigation.
>
> **How is the conversion from (assumingly) continuous feature vectors to integers done?**
>
> We use quantization to map continuous infinite values to a smaller set of discrete finite values. In our case, we use the implementation provided by Crypten to convert float numbers to 16-bit integers.

---

> > ### Author Response · Authors · 2022-08-02
> > **Responses to Reviewer BTzU**
> >
> > **Does the system need to share keys between the parties?**
> >
> > In our paper, we use the term “share” instead of “key” in order to remain consistent with existing literature [7, 16]. The “shares” only need to be distributed across parties for some operations in Shamir’s secret sharing schemes (e.g. multiplication and producing the final output).
> >
> > **Why during training (Eq 6) is finite field arithmetic not used?**
> >
> > All training in MPC is performed through finite field arithmetic. To indicate this, Eq. 2 defines F with finite field arithmetic.
> >
> > Thank you again for your helpful comments and please let us know if you have more questions.

---

### Official Review · Reviewer_t8m7 · 2022-07-14

**Rating:** 7
**Confidence:** 3
**Soundness:** 3 good
**Presentation:** 3 good
**Contribution:** 3 good

**Summary:**

- The paper proposes a framework for multiparty perception: on leverage multiparty computing techniques in a fully observable environments (with multiple cameras), such that the agent learns to take actions (e.g., driving direction) without observing the raw images from the multiple cameras.
- The core idea is that only encrypted ciphertexts extracted from the cameras are publicly exchanged between the parties, which are used to determine the actions.
- The approach is evaluated on an "obstacle world" dataset which will be released publicly. The goal of the agent is to reach a goal location while avoiding obstacles.
- Evaluation indicates that the performance of the proposed model reaches accuracies very close to the plaintext model (96.9% vs 97.1%).

**Questions:**

1. See concern 1: I would appreciate some clarification on agents vs. parties (since I think in MPC literature, generally agents = parties?)

**Limitations:**

Yes, the paper has a good discussion on limitations and societal impact.


**Strengths And Weaknesses:**

### Strengths
1. The problem tackled by the paper is well-motivated: while fully observable environments are highly useful for navigation, they incur a privacy cost. The approach suggests MPC can help achieving great performances with rigorous privacy guarantees.
2. The authors plan to publicly release the dataset which will be beneficial to the community.
3. The evaluation is rigorous and is accompanied with appropriate baselines and discussions.
4. The writing of the paper for the most part is clear and precise.

### Concerns

**1. Agent vs. "Party" vs. Camera views**
- The concern is to do with difference between agents and parties - are they the same? As in, is "two-party" case as shown in Figure 2 involves two agents? In which case, it appears that that first person view observed by each party would change making it unclear on how actions generalize between different agents with different goals.
- Alternatively, does one agent taking an action entail receiving encrypted ciphertexts from all other agents (each with their own first person view and computations)?

**2. (Nitpick) Some conclusions in Qualitative Analysis**
- I appreciate the authors presenting and discussing the success/failure examples of the approach.
- However, some conclusions drawn from the failure modes are not evident. For instance, take the conclusion L269 "the agent incorrectly reasons the goal location is behind the obstacle location" isn't entirely evident in Fig. 7 (middle row, left).

---

> ### Author Response · Authors · 2022-08-02
> **Responses to Reviewer t8m7**
>
> Thank you for your thorough review and important questions! We hope the responses below clarify the key points, and please let us know if there is any further information we can provide.
>
> **Agent vs. Party vs. Camera views**
>
> Thanks for pointing this out. We have improved the paper to clarify the terminologies (see result section line 216-220 and other minor edits throughout the paper), which we also summarize below.
>
> “Agent” refers to the vehicles or moving entities making a decision and taking an action. Our algorithm controls the navigation decisions of this vehicle/entity.
>
> “Party” refers to the number of stakeholders involved in the agents’ action decision making process. In other words, it refers to the number of parties performing secret sharing protocols in multi-party computation. In our case, one of the parties will always be the agent itself taking an action. The rest of the parties are created by the security cameras, and we explored two settings for how the security cameras are combined. In the first setting, all the security cameras are owned by one party, such as the government, so there are just two parties in total: the agent and all the other cameras (2-party computation). This assumes the security cameras trust each other. In the second setting, each security camera is owned by a different party, so there are many parties (n-party computation). This assumes the security cameras do not trust each other. The second setting is more computationally difficult to achieve, but we found our approach works well in both settings.
>
> “Camera views” refer to the first-person camera view owned by the agent and external security camera views owned by other parties like governments and companies. All the camera views together collectively construct a fully observable environment that will be crucial to effective navigation.
>
> Please let us know if this clarifies the terms, and we are happy to provide additional explanations if needed.
>
> **Is "two-party" case as shown in Figure 2 involves two agents?**
>
> The two-party case refers to an agent taking an action (party 1), and multiple camera views all owned by one entity (party 2, e.g. government with security cameras).
>
> **Alternatively, does one agent taking an action entail receiving encrypted ciphertexts from all other agents (each with their own first person view and computations)?**
>
> Yes, this is the case. Please let us know if we understand the question correctly. To be more precise, the agent taking an action receives encrypted ciphertexts from all other parties. Each party has one or multiple camera views and performs computation on encrypted ciphertext.
>
> **Conclusions drawn from the failure modes**
>
> Thanks for noticing this. As a clarification, the conclusion from L269 was referring to the two examples in the last row, not the middle row. You are absolutely right that the conclusion may not be entirely evident, and we decided to remove the sentence to keep the analysis more factual and neat.
>
> Thank you again for your helpful feedback and suggestions, and we are happy to answer any additional questions.

---

### Meta-Review · Area_Chair_4YYH · 2022-08-29

**Recommendation:** Accept
**Confidence:** Certain

**Metareview:**

*Summary*

-  The paper proposes a framework for multiparty perception: on leverage multiparty computing techniques in a fully observable environments (with multiple cameras), such that the agent learns to take actions (e.g., driving direction) without observing the raw images from the multiple cameras.
- The core idea is that only encrypted features extracted from the cameras are publicly exchanged between the parties, which are used to determine the actions.
- The approach is evaluated on an "obstacle world" dataset which will be released publicly. The goal of the agent is to reach a goal location while avoiding obstacles.
- Evaluation indicates that the performance of the proposed model reaches accuracy very close to the non-private model (96.9% vs 97.1%).

*Reviews*

- The paper received 3 reviews with final ratings: 7 (Accept), 6 (Weak Accept) and 4 (Borderline Reject).
- In general the reviewers found the paper to be well-motivated (mitigate privacy costs), well-written, with rigorous experiments and a dataset contribution.
- The reviewers raised some minor concerns regarding terminology and presentation, which have been resolved and are easily addressed in a camera-ready version.
- The more substantive concerns raised by reviewers were:
  - It might be possible to train a network to reconstruct camera images from the encrypted feature vectors. The authors' have convincingly responded to this concern by pointing to prior work and providing an additional experiment.
  - Would the method work in a real-world environment where there is a lot more variation between scenes and the map and cameras exhibit noise, occlusion and artifacts? The authors' argue that the paper is a first step towards this aim, which would constitute follow-up work.

*Decision*

- It's the view of the AC that the 'Borderline Reject' review (N4st) did not actually identify any substantive weaknesses in the paper that would warrant rejection. Reviewer N4st's main concern is 'limited technical contributions' which is convincingly rebutted by the authors. I'm swayed by the two positive reviews; the paper should be accepted.

**Award:**

No

---

### Decision · Program_Chairs · 2022-09-14

Accept